# Taking charge of eczema self-management: a qualitative interview study with young people with eczema

Kate Greenwell ![ORCID] ,[1] Daniela Ghio,[2,3] Ingrid Muller ![ORCID] ,[2] Amanda Roberts,[4] Abigail McNiven,[5] Sandra Lawton,[6] Miriam Santer[2]

► Prepublication history and additional materials for this paper are available online. To view these files, please visit the journal online (http://dx.doi.org/10.1136/bmjopen-2020-044005).

For numbered affiliations see end of article.

**Correspondence to**
Dr Kate Greenwell;
K.Greenwell@soton.ac.uk

## ABSTRACT

**Objectives** To explore young people's experiences of eczema self-management and interacting with health professionals.

**Design** Secondary qualitative data analysis of data sets from two semistructured interview studies. Data were analysed using inductive thematic analysis.

**Setting** Participants were recruited from the UK primary care, dermatology departments and a community-based sample (eg, patient representative groups, social media).

**Participants** Data included 28 interviews with young people with eczema aged 13–25 years (mean age=19.5 years; 20 female).

**Results** Although topical treatments were generally perceived as effective, young people expressed doubts about their long-term effectiveness, and concerns around the safety and an over-reliance on topical corticosteroids. Participants welcomed the opportunity to take an active role in their eczema management, but new roles and responsibilities also came with initial apprehension and challenges, including communicating their treatment concerns and preferences with health professionals, feeling unprepared for transition to an adult clinic and obtaining treatments. Decisions regarding whether to engage in behaviours that would exacerbate their eczema (eg, irritants/triggers, scratching) were influenced by young people's beliefs regarding negative consequences of these behaviours, and perceived control over the behaviour and its negative consequences.

**Conclusions** Behavioural change interventions must address the treatment concerns of young people and equip them with the knowledge, skills and confidence to take an active role in their own eczema management.

## INTRODUCTION

Eczema is a common skin condition that can have a significant effect on an individuals' quality of life, mainly due to persistent itching and pain, sleep disturbance and emotional distress.[1,2] Although eczema typically starts in infancy and epidemiology studies show that it typically improves or resolves by late childhood,[3,4] for many, symptoms can persist into adolescence and adulthood.[5] For most, eczema management focuses on the regular use of emollients to retain the skin's barrier

### Strengths and limitations of this study

► Our findings provided a more in-depth understanding of young people's unique experiences of eczema self-management and interactions with health professionals, which are largely not understood.

► Our in-depth data set included diverse views from young people across a wide age range and from different ethnic groups.

► Although researchers from the primary research were involved in the data analysis, secondary analysis can overlook the socio–cultural–political context under which the research was originally conducted.

► For one interview study, sections of the data set were to be published online, which meant that some participants may have been particularly motivated and may give socially desirable answers.

function; treating flare-ups using topical corticosteroids (TCS); identification and avoidance of irritants/triggers; and avoiding scratching, which can further exacerbate eczema symptoms.[6] For those with moderate and severe eczema, management may also involve use of topical calcineurin inhibitors, bandages or medicated dressings, systemic therapy (immunosuppressants) and phototherapy (light therapy).[6]

Self-management can be particularly challenging during adolescence and early adulthood. Young people must take on a more active role in their eczema management, a role that was previously the primary responsibility of their families. They experience new demands, including understanding and adhering to complex treatment regimens; financing, acquiring and testing effective medical treatments; interacting with health professionals; and negotiating healthcare systems. Such demands can serve to disrupt their health and well-being, and may lead to non-adherence and ineffective resource use.[7–9]

The unique self-management challenges and preferences of this group are largely not

BMJ

understood.[10] A recent systematic review found that most qualitative studies sampled young people with eczema alongside adults, children or people with other skin conditions, making it difficult to tease out the unique experiences of this group.[10] The current study is the third from our research team to explore young people's experiences of eczema and its management. Each study involved a secondary data analysis of a large qualitative data set derived from two primary studies exploring young people's experiences of eczema, but focused on a different aspect of these accounts. The first study explored perceptions about the nature of eczema (eg, as an episodic long-term condition) and how these perceptions related to their self-care and adaptation to eczema.[11] The second study explored young people's experiences of eczema-related symptoms (both visible and invisible to others) to determine their psychosocial needs.[12] The current study aims to explore young people's experiences of eczema self-management and interacting with health professionals. This study will go beyond previous qualitative studies with young people[9] by exploring perceived challenges around other self-management behaviours (ie, avoiding irritants and triggers, reducing scratching), as well as topical treatments and, crucially, their experiences of taking an active role in eczema management.

## METHODS
### Design and data collection
The study design was a secondary analysis of qualitative interview data, drawing on two data sources. Although both studies had different aims, both used interview questions relevant to this study aim, exploring experiences of eczema treatments and management, and interactions with health professionals.

### SKINS project
This project aimed to explore young people's experiences of living with common skin conditions (eczema, acne, psoriasis, alopecia) and share these on healthtalk.org. Participants were recruited from health settings, patient representative groups, educational institutions and social media. Maximum variation sampling was used to recruit a range of demographics.[13] Twenty-four of the 97 interviews were with young people (aged 17–25 years) with eczema and all but one consented for their data to be used for this secondary analysis. Semistructured interviews were carried out by AM (qualitative health researcher) between October 2014 and December 2015. They took place at the participants' homes or alternative meeting places, and lasted up to 2 hours. Interview topics included views about eczema treatment and management; experiences of healthcare; and advice for others with eczema and health professionals (online supplemental file 1). Interviews were either audio or video recorded and transcribed verbatim, and checked by participants for accuracy. All three of our secondary data analysis studies drew on this data source.

### Eczema care online project
As part of a wider project to develop an online intervention for young people with eczema, a survey and semi-structured interviews were carried out with this group. Interviews recruited younger adolescents (13–16 years) who were not captured in the SKINS sample. Sixteen general practitioner (GP) practices and three hospitals across Wessex provided an invitation letter, participant information sheet and survey to 1123 young people with eczema (aged 13–25 years). Fifty-five young people agreed to take part in the interviews, 12 were 13–16 years old. Participants were purposively sampled to ensure a range of ages and a balance of genders. Five young people took part in the interviews. Those who did not take part in the interview either could not be contacted by telephone or were not selected for purposive sampling. This data source was also used in one of our other secondary data analysis studies.[12]

DG (expertise in paediatric psychology and qualitative research) interviewed participants at their homes during March–May 2018 lasting 22–50 min. Participants' parents were present for three of the interviews. Interview topics included views and experiences of eczema, treatments, self-management, and information and support for eczema (online supplemental file 2). Interviews were audio recorded and transcribed verbatim.

Of the 28 participants overall, 20 were female and 8 male. Participants had a mean age of 19.5 years and had eczema for a period of 8 months to all their life (table 1).

### Patient and public involvement
Our patient collaborator (AR) has eczema and children with eczema, and helps run a support group for carers of children with eczema. She was a co-applicant on the research grant application, helping to identify the research topic and develop research questions. She attended regular project meetings, discussed and provided feedback on our interpretations of the findings, and provided feedback on the eczema care online (ECO) study materials and this manuscript. She continues to help us to disseminate our research findings among her wide-reaching patient networks and via social media. Two young people reviewed the ECO participant information sheet to check comprehension. For the SKINS project, eight patient and public involvement (PPI) representatives (including young people with eczema, parents and skin charity members) attended advisory group meetings and reviewed the interview schedule and summaries of the findings.

### Data analysis
Both data sets were analysed together by following the six stages of Braun and Clarke's inductive thematic analysis[14] and data handling was facilitated using NVivo V.12 Pro. DG carried out initial coding on the entire SKINS data set, familiarising herself with the data and generating initial codes that represented the various topics present across the data (eg, barriers to emollients, experiences

**Table 1** Demographics of participants from the two data sources

|  | Frequency | % |
|---|---|---|
| **SKINS project** | | |
| Gender | | |
| Male | 6 | 26 |
| Female | 17 | 74 |
| Age (years) | | |
| 16–18 | 4 | 17 |
| 19–21 | 10 | 43 |
| 22–24 | 9 | 39 |
| Ethnicity (self-identified) | | |
| White British | 13 | 57 |
| British Indian | 3 | 13 |
| Bangladeshi | 2 | 9 |
| Pakistani | 2 | 9 |
| Indian | 1 | 4 |
| Mixed—Black Caribbean and White | 1 | 4 |
| Mixed—White and Algerian | 1 | 4 |
| White Hungarian | 1 | 4 |
| Duration of condition | | |
| All life (diagnosis before first birthday) | 16 | 70 |
| 11–21 years (diagnosis in toddler years) | 4 | 17 |
| 5–7 years (diagnosis in adolescence years) | 2 | 9 |
| Up to 1 year (diagnosis in young adult years) | 1 | 4 |
| **Eczema care online project** | | |
| Gender | | |
| Male | 2 | 40 |
| Female | 3 | 60 |
| Age (years) | | |
| 13–14 | 2 | 40 |
| 15–16 | 3 | 60 |
| Duration of condition | | |
| All life (diagnosis before first birthday) | 4 | 80 |
| 3 years (diagnosis in adolescence years) | 1 | 20 |
| Eczema severity (self-identified) | | |
| Moderate | 3 | 60 |
| Moderate/severe | 1 | 20 |
| Mild | 1 | 20 |

with health professionals). A coding manual was created, including descriptions of the codes and example quotes, and DG discussed this with KG, IM (both health psychologists experienced in qualitative research) and MS (GP experienced in qualitative research). These three researchers used the manual to code two transcripts, and changes to the coding manual were made following discussions. The revised coding manual was discussed with a PPI representative (AR) and the interviewer for

the SKINS project (AM). It was agreed that the findings would be reported across three separate studies (the other two are reported elsewhere[11 12]).

For this study, KG reviewed the initial codes deemed relevant to the study aim, redefined them in line with this aim, searched for, reviewed and defined themes, and created a coding manual for this study. KG then familiarised herself with the ECO data, and coded the data, identifying any data that deviated from the SKINS data (eg, any new codes or experiences) in the coding manual. Disconfirming cases were identified throughout the entire data set. Writing up the thematic analysis in this manuscript helped finalise the analysis, enabling the final themes to be reviewed and agreed by all coauthors. Pseudonyms were given to all participants.

An important quality criterion in qualitative research is the adequacy of the sample size. This can be assessed by consideration of a study's 'information power'[15] and whether data saturation was reached.[16] Drawing on the guidelines on information power in qualitative interview studies,[15] our sample size for the secondary data analysis (n=28) was deemed adequate given the specificity of the study aim (self-management) and target group (young people), the large amount of data (in particular from the SKINS interviews, which lasted up to 2 hours) and the likely high quality of dialogue from using experienced qualitative postdoctoral researchers. It was also comparable with other qualitative interview studies exploring experiences among people with eczema.[10] When analysing the SKINS data, we felt that data saturation was reached as the later interviews did not identify any additional codes or diversely different experiences or views.[16] Although the analysis of the ECO project data did not identify any additional codes (ie, code saturation was reached[16]), the analysts felt that additional interviews with this age group were needed to be confident that the issues facing this group are fully understood (ie, meaning saturation was not reached for this group[16]).

## RESULTS

The five themes and the key findings under each theme are summarised in figure 1.

### Beliefs about the effectiveness of topical treatments

Generally, young people believed that their topical treatments (emollients and TCS, topical calcineurin inhibitors) were effective. However, many described experiences whereby certain treatments had stopped working with time or caused new side effects (eg, burning). This led them to change emollient or TCS, apply increasingly more treatment to get the same effect or try a stronger TCS. Some participants attributed the reduced effectiveness of the treatments over time to their skin adapting to treatments (mainly emollients).

> I can use one moisturiser for six months, a year, and then my skin will get too used to it and I'll have to

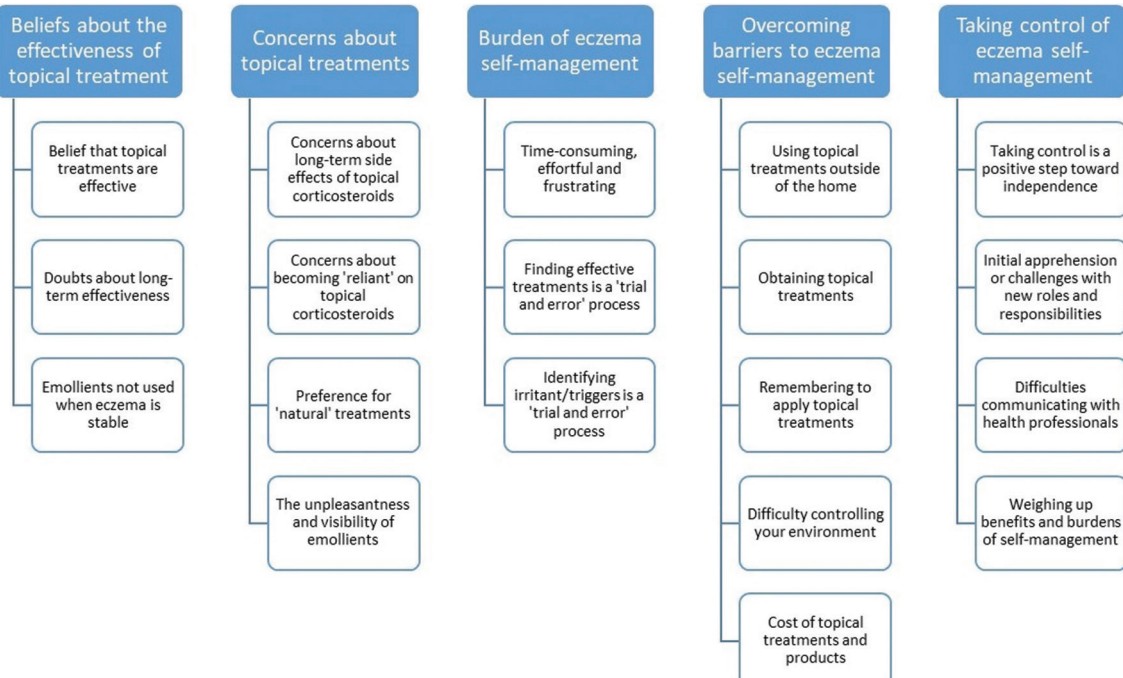

**Figure 1** Visual representation of themes and key findings.

keep putting more on. Then I'll end up having to switch to a different kind of moisturiser just so I can get it under control again. (Cassie, female, 15 years old, eczema since birth, ECO project)

Switching around [emollients], generally, does make a difference. (Mali, male, 23 years old, eczema for 20 years, SKINS project)

A minority reported how they had tried certain topical treatments (mainly TCS) that made their eczema worse, and a few did not find topical treatments made a difference to their eczema. A few participants stopped using emollients when their eczema was better or used them only when their eczema was problematic: "I'm sort of taking the 'if it isn't broke don't fix it' route with my skin." (Eevi, female, 22 years old, eczema since birth, SKINS project).

### Concerns about topical treatments
Many young people expressed a general caution or hesitancy around using TCS, listing side effects (eg, burning) or long-term effects they had experienced or had heard about (eg, skin thinning, skin ageing, impairing immune system). For some, this cautiousness was about acknowledging the importance of using TCS correctly and avoiding excessive and unnecessary use, whereas others actively avoided using TCS or only used them when their eczema was very bad. Two participants expressed concerns that TCS might cause eczema to 'spread' or move around the body:

I feel that steroids sort of suppress it [eczema] and it crops up somewhere else...I want to get rid of the whole thing, not just have this internal thing inside me that will just crop up when and where because I'm

not using something to suppress it. (Iris, female, 20 years old, eczema for 16 years, SKINS project)

Some participants were worried about becoming 'reliant' on TCS, having to use more, or increasingly stronger, TCS or having to use them indefinitely. Many of their TCS concerns had originated from stories about potential side effects they had read or heard from others.

[I] met a girl who said that they [TCS] had the capacity to change your sexuality...so since then I've been really nervous [to use them]. (Hannah, female, 20 years old, eczema for 16 years, SKINS project)

[I] read about a lady...she used steroids for such a long time that it's definitely sort of had an adverse sort of reaction to her body...her hair's thinned and her...nails are brittle. (Dua, female, 21 years old, eczema for 20–21 years, SKINS project)

Such concerns about side effects led some participants to switch to using more 'natural' products to help eczema, sometimes against or at least without consulting medical advice. Natural (or homemade) products were viewed as 'healthy', described as containing no or fewer chemicals, and less likely to cause side effects.

I wanted to stay away from that [potent TCS] because...I like naturally occurring products...my dermatologist...wasn't too happy for me to go down the route without steroids I think because...it does have a really high success rate. I won't deny that. (Iris, female, 20 years old, eczema for 16 years, SKINS project)

Two participants described how they had experienced side effects (red, heated and stinging skin) from using a topical calcineurin inhibitor.

Compared with TCS, young people expressed fewer concerns regarding the safety of emollients. A few participants explained how emollients can 'block' their skin and one expressed concerns about the flammability of emollients containing paraffin.

Participants found that topical treatments, specifically emollients that are applied regularly in large amounts, can be unpleasant in texture and/or smell, can make it difficult to use certain object (such as a pen), can rub off onto other items, such as clothes or bedding, can make them feel hot, sweaty or 'dirty', and can sting when applied. Some avoided applying creams in public as they did not want others to 'find out' they had eczema. They worried that others could smell their treatments, or that people would not want to touch them (eg, partners holding their hand) if they had creams on. However, some explained how such worries reduced as they got older and started to care less about the others' opinions.

> My colleagues often see me moisturising at my desk. They don't mention it [laughs] And if people think it's a bit weird, like I feel bad, but also like it's so uncomfortable when you can't moisturise. (Willow, female, 23 years old, eczema since birth, SKINS project)

### Burden of eczema self-management

Participants explained how burdensome eczema self-management could be. Applying topical treatments, identifying and avoiding irritants/triggers (especially as there are potentially so many), resisting the urge to scratch, and the 'trial and error' process of finding effective and acceptable (eg, does not sting) treatments were described to be time-consuming, effortful, frustrating, and sometimes expensive.

> It was quite a repetitive experience of just back and forth and back and forth. You'd explain [to the doctor], 'oh okay well it's not working', like 'okay let's, let's higher the dosage. Or let's try and find some steroids in', it just would not work. (Raashid, male, 24 years, eczema since birth, SKINS project)

> Some days you just can't be bothered. If you're out with friends, you know it's time to put on your cream; you would ignore it and during that night, I'd sometimes suffer the consequences…and I was scratching and itching all the time. (Qamar, male, 18 years old, eczema for 11 years, SKINS project)

As can be seen in the above quote, treatment burdens led some young people to skip treatments, especially when they had little time or energy.

### Overcoming barriers to eczema self-management

Young people listed several practical barriers to eczema self-management and shared strategies they found helpful for overcoming these. Participants found it challenging to use their topical treatments (mainly emollients) when outside of the home. Specifically, it was difficult to use creams during class or take them when travelling abroad on a plane (due to luggage restrictions); or apply them if they worked in a public-facing job (eg, retail or waiter). Other barriers to accessing treatments included running out of treatments, being unable to get a doctor's appointment to request more treatment (especially if they are living away from home and need to register with a GP) or treatment brands becoming discontinued. One younger participant from the ECO project reported that he found it hard to know how much emollient to apply. Participants found it helpful to keep sample-sized emollients or emollients decanted into smaller containers with them (eg, in their bag) or around the house, to apply emollient before and after they leave the house, to stock up on treatments to avoid running out and to be able to buy treatments without a prescription.

Young people explained how it could be difficult to remember to apply topical treatments and emphasised the importance of having a routine for applying treatments (especially their emollients), applying their creams at the same time each day or making time to apply the creams: "The biggest thing for fighting eczema is routine…do something that you think is working and do it regularly… don't be disorganised with it." (Zaahira, female, 21 years old, eczema since birth, SKINS project).

Although young people acknowledged the importance of avoiding irritants and triggers, they explained that it could be difficult to control your environment, for example, staying at a friend's house, or in shared accommodation or an office environment. There were also external factors that they perceived to be unavoidable or outside of their control, such as the weather or stress.

Many participants commented on the cost of topical treatments, and cosmetics or washing products (eg, shower gels, laundry detergents) that do not cause flare-ups. When they were younger, participants' treatments were free on the National Health Service, but many had to pay for their treatments from the age of 18 years. This cost prevented some from getting more treatment or led them to buy non-prescription treatments instead (eg, over-the-counter medications). Participants particularly resented the cost when buying treatments or products that were ineffective or made their eczema worse.

> I don't want to go to the doctors and I don't want to get more prescriptions because it just costs so much… once I had to fork out sort of about £36 worth of medications just for one particularly bad flare-up…and then if it doesn't work you're like, 'What, I spent £40 on nothing'. (Dua, female, 21 years old, eczema for 20–21 years, SKINS project)

Some participants explained how they were still eligible for free prescriptions or obtained prescription prepayment certificates, which helped with their prescription costs.

## Taking control of eczema self-management

Young people shared their experiences of taking control of their own eczema management. They explained how, when they were a child, their parents did the majority of their eczema management. As they became a teenager, they started taking greater responsibility for, and playing a more active role in, their eczema care. Generally, this transition was welcomed by participants and seen as a positive step towards becoming independent, a goal that was important to them.

> As a child, I wasn't managing [eczema] at all, I'd let my parents do it [applying cream] whenever they felt like it needed being doing. But I wouldn't be applying my cream…And as [a] teen, I realised that it was my problem and that I should be doing it and obviously as a teen you want your independence. You don't want your parents like stripping you off and covering you in bandages and cream. (Jamila, female, 20 years old, eczema since birth, SKINS project)

> My mum has always wanted to come with me to my appointments when I was older. But I've always said, 'I want to do it on my own because I want to speak to my doctor.' Cos [laughs] she had a habit when I was younger of coming with me to the doctor and…deciding what was being done for me. (Iris, female, 20 years old, eczema for 16 years, SKINS project)

Like Jamila in the quote above, young people's drive for independence was prompted by a realisation that eczema management was their 'problem' (rather than their parents' or health professionals') and they would have to live with the consequences of poorly managed eczema. For some, independence was important for maintaining privacy and avoiding their parents seeing them as vulnerable or distressed.

In contrast, some young people highlighted initial apprehension or challenges with these new roles and responsibilities, including moving to an adult clinic, having to do more things for themselves and taking an active role in their medical consultations.

> I'm not really looking forward to that because…they don't tell you what to do apparently in the adult clinic…you have to be by yourself in the adult clinic so you have to…pay more attention and they're not going to help me out as much as the children's nurse would. (Adi, male, 17 years old, eczema for 6.5–7 years, SKINS project)

Although young people were keen to take an active role, some found it intimidating to talk to health professionals or question their suggested treatments.

> When I go to the doctors I feel like they're the person in charge so I can't really sort of be like, you know, 'I know that this suits me best'…I think you do get intimidated…they sort of just give you like, you know that doctor tone, they're just a little bit better than you…and you're like, 'OK I guess you guys are right but [pause]'. (Dua, female, 21 years old, eczema for 20–21 years, SKINS project)

Many participants experienced times when their doctors or pharmacists were reluctant to prescribe their preferred treatments or refer them to a dermatologist. Conversely, some participants shared positive experiences of obtaining their preferred treatments from their health professionals and praised them for listening to them and their willingness to try different treatments. Willow (female, 23 years old, eczema since birth, SKINS project) explained how she asked her GP if she could be given a topical calcineurin inhibitor after a friend with eczema had recommended it. Although her GP prescribed it for her, she felt frustrated that she had to find out about this herself and the GP had not recommended it to her earlier. Young people stressed the importance of doing your own research around treatments (eg, looking online), being 'confident' or 'insistent' when communicating needs and preferences to health professionals, and writing a list of things to discuss beforehand.

Despite their changing role, the young people's families were still an important source of support for them: driving them to appointments, providing advice on eczema management, helping them apply topical treatments in difficult-to-reach areas, helping them understand what the doctor said and speaking on their behalf at medical appointments when they felt unable to. This support was particularly important for early to mid-adolescents whose parents were still involved in their eczema management.

> [When I apply treatments depends on] how I'm feeling at the time. So if Mum's at the point of, 'Oh [participant name], go put your creams on', I'm like, 'Okay, I'll go put my creams on' and I'll do it. Then I'll come downstairs and she's double checking, 'Oh have you done your creams?' At times she'd say it and I'll say, 'Yes, I'll go do it and then I'll completely forget,' and that [is] pretty much the days where I would not do them. (Ben, male, 14 years old, eczema for 6 months, ECO project)

When deciding whether to engage in specific self-management behaviours, participants considered whether doing so was worth the potential impact it would inevitably have on their eczema. For example, decisions around avoiding irritants/triggers were influenced by how difficult it was to avoid (eg, if it was their favourite food, if everyone around them is drinking alcohol or eating a certain food), their belief about how much it would make their eczema worse and whether any action could be taken to minimise these consequences (eg, by applying emollients or taking antihistamines afterwards).

> If I cut out lactose I don't think that it would make that much of a difference, so I might as well just eat the cheese and be happy. (Eevi, female, 22 years old, eczema since birth, SKINS project)

I'd probably just carry on doing them [sports] and just sort of live with the consequences so, as long as you…go and have a shower afterwards and…cool down…if I'm having fun doing something then I'm not going to stop just cos of eczema. (Lanie, female, 24 years old, eczema for 20+ years, SKINS project)

Participants were aware of the negative consequences of scratching, however, many found it difficult to avoid scratching completely. This was because a lot of their scratching was done subconsciously, it was a habit, many of the self-management strategies (eg, tapping) were ineffective or the relief of scratching was rewarding.

I'm terrible for scratching. I cannot stop myself. It's like an addiction cos it relieves such a horrible sensation all round my body. (Iris, female, 20 years old, eczema for 16 years, SKINS project).

I normally try to stop, like stop myself from doing it, but it gets to the point where it's just all over, all over my neck and I just have to scratch it. (Amelia, female, 15 years old, eczema for 12 years, ECO project)

## DISCUSSION

Our findings provided a more in-depth understanding of young people's unique experiences of eczema self-management. Specifically, the challenges they face when learning how to take an active role in their self-management: an issue that has not been explored in previous research. Although the new roles and responsibilities around eczema management were generally welcomed by young people, they also brought initial apprehension and challenges. These findings mirrored previous research highlighting that young people often feel unprepared when transitioning from paediatric to adult care.[17 18]

Participants in this study found it hard to communicate their needs and treatment concerns to health professionals, which meant that, at times, they left consultations feeling like they were not listened to or were dissatisfied with the recommended course of action. Our previous qualitative study with this group found that young people with eczema have a strong need to feel understood and 'taken seriously' by their health professionals.[12] The more confident and competent young people feel in communicating their needs to health professionals, the more understood and validated they are likely to feel. There is a need for health professionals to acknowledge and encourage young people's desire to become active contributors to their own healthcare, treating them as an equal partner in their care, while recognising that this new role may be initially daunting for some.[19]

Our study explored how young people decide whether to engage in behaviours that would exacerbate their eczema (eg, irritants/triggers, scratching). Previous research has focused predominantly on the perceived benefits and costs of topical treatments.[10] Young people's

decisions were influenced by their beliefs regarding the negative consequences of the behaviours, and their perceived control over the behaviour and its negative consequences. People with eczema have a desire to live a 'normal' life and not allow their eczema management to impact on their everyday lives.[10 12] Behavioural change interventions can help promote self-management and, as argued by the Self-determination Theory, such interventions are more effective if they enhance an individual's need for autonomy and minimise conscious effort and lifestyle disruption.[20–22] It is important to support young people to decide which of these behaviours can be avoided and how to minimise the consequences of unavoidable irritants/triggers, focusing on what is positive and possible.[19]

The Burden of Treatment Theory[23] provides a useful framework for interpreting our study findings and making implications for practice. It argues that health interventions can minimise treatment burdens by building an individual's cognitive and material capacity to perform self-management tasks. In the context of our findings, this may involve building a young person's knowledge and skills required for making informed treatment decisions and weighing up the benefits and costs of self-management behaviours. For example, the financial cost of treatments is a common barrier among young people.[10] To build their material capacity, health professionals should acknowledge young people's concerns around the costs and educate them regarding any financial support that is available to them. The Burden of Treatment Theory argues that interventions should build and strengthen 'relational networks' (ie, social networks, health professionals) around patients, and equip them to effectively navigate healthcare systems. Eczema interventions should build young people's confidence and communication skills and support them to make decisions, so they can direct their own care and feel prepared when transitioning to adult clinics.[19]

There are several limitations to this study. Secondary qualitative analysis has been criticised for not considering the socio–cultural–political context under which the primary research was conducted.[24] This issue is particularly salient for this study as the two sets of primary data were collected at two different time points. However, to minimise such decontextualisation, avoid any misinterpreted data and ensure the final interpretations were representative of both data sets, the interviewers of both data sets were coauthors (DG, AM).

Although both primary studies explored the topics relevant to the aims of the current study, these studies were conducted for different purposes (ie, to inform two different health interventions), which may have introduced differences in the data. For example, sections of the transcriptions and, with participant consent, audio and video clips from the SKINS interviews were to be published online. Therefore, compared with the ECO study participants, SKINS participants may have been more motivated and more likely to give socially desirable

answers. Despite this, our analysis highlighted that there was a great deal of convergence between the two data sets and our use of disconfirming case analysis ensured that any differences were identified and acknowledged. Our findings are also congruent with other data sets.[9 10 17 18]

As this was a secondary data analysis, it was not possible to carry out additional interviews in order to have full confidence that data saturation was reached. Moreover, the sample included only eight male. Additional interviews with adolescents aged 13–16 years and male with eczema may have enabled us to explore any differences between the age groups and between genders in more depth.

In conclusion, our findings highlight key self-management challenges faced by young people with eczema. Behavioural change interventions must address the treatment concerns of this group and equip them with the knowledge, skills and confidence to take an active role in their eczema management.

**Author affiliations**
[1]Centre for Clinical and Community Applications of Health Psychology, Faculty of Environmental and Life Sciences, University of Southampton, Southampton, UK
[2]Primary Care, Population Science and Medical Education, Faculty of Medicine, University of Southampton, Southampton, UK
[3]Psychology, School of Health and Society, University of Salford, Manchester, UK
[4]Centre of Evidence-Based Dermatology, School of Medicine, University of Nottingham, Nottingham, UK
[5]Nuffield Department of Primary Care Health Sciences, University of Oxford, Radcliffe Observatory Quarter, Woodstock Road, Oxford, OX2 6GG, UK
[6]Dermatology Department, The Rotherham NHS Foundation Trust, Rotherham, UK

**Acknowledgements** We are grateful to the Medical Sociology and Health Experiences Research Group (University of Oxford) and Healthtalk.org for data sharing, and the young people who took part in this research.

**Contributors** KG, DG, IM, AR, AM, SL and MS were involved in designing and planning the study. DG and AM were responsible for recruitment and data collection. KG and DG led on the data analysis, with support from MS and IM. AR (PPI representative) and AM reviewed and provided feedback on the analysis and interpretations of the study findings. KG drafted the manuscript, with support from DG, MS and IM. MS, DG, IM, AR, AM and SL critically reviewed the manuscript, contributing important intellectual content and approved the final manuscript.

**Funding** This study was funded by the National Institute for Health Research (NIHR) Programme Grants for Applied Research (RP-PG-0216-20007). Data collection for the SKINS project was funded by NIHR under its Research for Patient Benefit scheme (PB-PG-0213-30006).

**Disclaimer** This manuscript presents independent research funded by the National Institute for Health Research (NIHR). The views expressed are those of the authors and not necessarily those of the NHS, the NIHR or the Department of Health and Social Care.

**Competing interests** None declared.

**Patient consent for publication** Not required.

**Ethics approval** The SKINS project was approved by Berkshire NRES Committee South Central, and the ECO data collection and both analyses were approved by Wales REC 7 Ethics Committee (REC 17/WA/0329).

**Provenance and peer review** Not commissioned; externally peer reviewed.

**Data availability statement** The data that support the findings of this study are available from the corresponding author upon reasonable request.

**ORCID iDs**
Kate Greenwell http://orcid.org/0000-0002-3662-1488
Ingrid Muller http://orcid.org/0000-0001-9341-6133

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
