## [Reviewer comments · BMJ Open]

ARTICLE DETAILS

TITLE (PROVISIONAL)	Taking charge of eczema self-management: a qualitative interview study with young people with eczema
AUTHORS	Greenwell, Kate; Ghio, Daniela; Muller, Ingrid; Roberts, Amanda; McNiven, Abigail; Lawton, Sandra; Santer, Miriam

VERSION 1 – REVIEW

REVIEWER	Dr Simon Tso Jephson Dermatology Centre, South Warwickshire NHS Foundation Trust, Warwick, CV34 5BW, United Kingdom.
REVIEW RETURNED	09-Oct-2020

GENERAL COMMENTS	Reviewer experience Thank you for inviting me to review this paper. I am a NHS consultant dermatologist with qualitative research training and experience, and I do not have any conflict of interests to declare. Overview I read the manuscript with interests. It is a well written and easy to follow paper. This is a secondary analysis of interview data gathered from two sources: the SKINS project (interviews took place between 2014-2015) and the ECO project (interviews took place in 2018). The authors explained the SKINS and the ECO projects have diverging project aims. The authors helpfully and rightly declared in the manuscript that this reported 'study is the third from their research team to explore young people's experiences of eczema and its management. The previous two studies focused on young people's adaptation to eczema and its psychosocial impact.' The authors referenced these two papers as Ref 3 and 9, which I have subsequently read in detail. It is my opinion that the relationship between this manuscript under review and the work reported in Ref 3 and 9 could be more clearly defined and declared by the authors (please see major comments 1 for full details about my major concerns). The methods section provided a useful account of the background and nature of the SKINS project and the ECO project. The authors briefly described the data was analysed using thematic analysis (see major comment 3). It is excellent that there was good quality patient and public involvement in the two projects and in this reported study. The authors reported five emerging themes. These themes were descriptive in nature but highly relevant to the young person's eczema experience. The discussion section is relatively well written and attempted to provide a deeper analysis of the emerging themes and their relevant to the patient journey. The study limitation was briefly discussed (see major comment 4). Major Comments This is a well written paper and I can relate the study findings to my professional experience caring for eczema patients from across all age groups. However, I have major concerns about how the study methods was reported and would be grateful for the authors to consider the following points
---

Major Comment 1

Please accept my apologies if I am mistaken, but based on the information reported in this manuscript, in Ref 3 and Ref 9 (shown below in bullet points), I am given the impression that all three papers drew data from exactly the same 23 interview data sets from the SKINS project; and this manuscript and Ref 9 further drew data from the same 5 interview data sets from the ECO project. However, all 3 papers appeared to report different aspect of young persons' accounts of their eczema experience.

- This submitted manuscript reported data source is from SKINS project (23 interviews with participants aged 17-25 years) and ECO project (5 interviews with participants aged 13-16 years)
- Ref 3 (Ghio et al, BJD, 2020) reported data source is from Healthtalk.org (SKINS project) involving 23 interviews with young people with eczema, of which 17 were female and 6 male, ranging from 17-25 years old.
- Ref 9 (Ghio et al, Br J Health Psychol, 2020) reported data source is from SKINS project (involving 24 young people with eczema aged 17-25 years old; one interviewee did not consent for secondary analysis i.e. 23 interviews) and ECO project (5 interviews with adolescent 13-16 years)

The benefits of clearly stating the relationship between all three papers and where the data was drawn from, will provide much needed clarity to readers about the study methodology and it is my opinion that all three papers should be read together in order to fully appreciate the young person's eczema experience. Analysing the same data with the same method, but breaking the data up and report findings across three different journals without clearly stating their relationship has reduced the impact of the work reported here.

Major Comment 2

It will be very useful for readers to understand the rationale or the author's reflexivity behind why did the authors pair up the SKINS project and ECO project data to be analysed together. Apart from having project participants from the young person age range, these two projects have diverging project aims and interview schedules, and carried out 3 years apart. Out of all the quotations presented in this submitted manuscript, only 1 quotations was from a ECO project participant. It will be useful for the authors to further reflect on how coding of data from ECO project impacts on coding of SKINS project data, and vice versa, as well as their their impact on the development of themes.

Major Comment 3

It is disappointing to see the data analysis method described in page 9 is so brief and does not adequately explain how the author's thematic analysis was performed, how was the data synthesised and whether the authors followed Braun and Clarke's six step thematic analysis or performed a variation of it. While the SKINS project coding has benefited from member checking, the authors stated KG coded the ECO data, and updated the coding manual accordingly – suggesting the ECO data did not benefit from member checking. Furthermore, Ref 3 and 9 provided a much fuller account of the study methods – the authors could have referenced those papers for full details of their data analysis method performed in relation to the work reported in this manuscript if they did not wish to provide the full details.

Major Comment 4

Page 19 study limitations

The limitations of this study has not been adequately discussed. The sample size of 28 interviews is at best a modest sample and data saturation was not achieved. The authors should also mention or reference the limitations they stated in Ref 3 and Ref 9 (due to

reasons already mentioned under Major Comment 1).

Below are my minor comments

Minor comment 1

Introduction page 5 line 16 stated 'Eczema management focuses on the regular use of emollients to retain the skin's barrier function; treating flare-ups using topical corticosteroids (TSC); identification and avoidance of irritants/triggers; and avoiding scratching, which can further exacerbate eczema symptoms.[4]'

Comments: This statement oversimplified the management of eczema, in contrast to the study findings that eczema treatment is complex for young people. Also, this is not a full description on the management of eczema. While the author's statement above is relevant to all severity of eczema, however, as the authors referenced the NICE guideline (Ref 4) they should consider further mentioning topical calcineurin inhibitors, bandages, systemic therapy and phototherapy for moderate and severe eczema.

Minor comment 2

Results section page 9-10 'This led to them to change emollients or TCS, apply increasingly more treatments to get the same effect, try a stronger TCS, or try a different type of treatment (eg. Phototherapy, immunosuppressants).

Comment: Dermatologists always make the effort to minimise TCS skin thinning risks by introducing steroid sparing agents like topical calcineurin inhibitors whenever indicated. Primary care has very good awareness of these steroid sparing agents and often initiated them prior to secondary care referral. Many young people that I care for with eczema can recall the name of the topical calcineurin inhibitor and know how to use it. Thus, I was very surprised that topical calcineurin inhibitors (Elidel cream and Protopic ointment) was not mentioned in the study findings at all. To me as a dermatologist, it is a significant finding that there is a lack of mentioning/awareness of topical calcineurin inhibitors amongst young persons which is second line treatment for eczema (before considering phototherapy or systemic immunosuppressants). I noted the authors' COREQ statement stated the study did not aim for saturation of data.

Minor points 3

Introduction page 5 line 10 stated

Although eczema typically starts in infancy and is expected to resolve by late childhood, for many, symptoms can persist into adolescence and adulthood.[3]

Comments: the sentence appeared ambiguous. In practice, roughly a quarter of children with eczema may find their eczema persist through to adulthood. Thus, I would be grateful if you could clarify the phrase 'expected to resolve by late childhood' by whom? Is it parents, the young person, the clinician or the society?

Minor Comment 4

For all quotations, could you kindly add in whether the quote was from a participant in the SKINS or ECO project please.

Minor Comment 5

Regarding Table 1 in this manuscript. It looks the same as the Table 1 in Ref 9. Consider adding a reference please.

Minor Comment 6

Reference 9 has been published and the authors should update this

	reference Minor Comment 7 The supplementary files 1 and 2 (Interview schedule for SKINS project and ECO Project are almost the same in this submitted manuscript and in Ref 9, but please pay attention to the differences in Supplementary 2 Interview Schedule – Eczema Care Online Project (see below) and make the necessary corrections as the interview schedules should be exactly the same. In this submitted version supplementary file 2, page 4 ends with That is all really useful, thank you. Is there anything that we haven't talked about that you would like to add? In Ref 9 supplementary file 2, page 4 ends with Views about wording – extracts Show the extract – That is all really useful, thank you. Is there anything that we haven't talked about that you would like to add? Thank you so much
--	---

REVIEWER	Belinda Sheary Royal Randwick Medical Centre Australia
REVIEW RETURNED	13-Oct-2020

GENERAL COMMENTS	Thank you for this interesting paper. 1. In the introduction TSC is written instead of TCS 2. on page 6, third paragraph, the wording of "from 8 months to all their life" was a bit unclear, I initially read it as from 8 months of age instead of 8 months in total. Could be clearer if written something like "had eczema for a period of 8 months to all their life" 3. there is inconsistency with TCS and TCSs both being used for the plural "topical corticosteroids" throughout the paper - eg in the introduction and last paragraph of page 8
--

VERSION 1 – AUTHOR RESPONSE

Reviewer 1 comments:

- Please accept my apologies if I am mistaken, but based on the information reported in this manuscript, in Ref 3 and Ref 9 (shown below in bullet points), I am given the impression that all three papers drew data from exactly the same 23 interview data sets from the SKINS project; and this manuscript and Ref 9 further drew data from the same 5 interview data sets from the ECO project. However, all 3 papers appeared to report different aspect of young persons' accounts of their eczema experience.**

- This submitted manuscript reported data source is from SKINS project (23 interviews with participants aged 17-25 years) and ECO project (5 interviews with participants aged 13-16 years)**
- Ref 3 (Ghio et al, BJD, 2020) reported data source is from Healthtalk.org (SKINS project) involving 23 interviews with young people with eczema, of which 17 were female and 6 male, ranging from 17-25 years old.**
- Ref 9 (Ghio et al, Br J Health Psychol, 2020) reported data source is from**

SKINS project (involving 24 young people with eczema aged 17-25 years old; one interviewee did not consent for secondary analysis i.e. 23 interviews) and ECO project (5 interviews with adolescent 13-16 years)

The benefits of clearly stating the relationship between all three papers and where the data was drawn from, will provide much needed clarity to readers about the study methodology and it is my opinion that all three papers should be read together in order to fully appreciate the young person's eczema experience. Analysing the same data with the same method, but breaking the data up and report findings across three different journals without clearly stating their relationship has reduced the impact of the work reported here.

RESPONSE: We agree that clarifying the relationship between all three papers would strengthen this manuscript, therefore, we have added the following detail to the introduction:

“Each study involved a secondary data analysis of a large qualitative data set derived from two primary studies exploring young people’s experiences of eczema, but focused on a different aspect of these accounts. The first study explored perceptions about the nature of eczema (e.g. as an episodic long-term condition) and how these perceptions related to their self-care and adaptation to eczema.[11] The second study explored young people’s experiences of eczema-related symptoms (both visible and invisible to others) to determine their psychosocial needs.[12] The current study aims to explore young people’s experiences of eczema self-management and interacting with health professionals.”

We have also clarified in the methods what data was used in the other two secondary data analysis studies.

“All three of our secondary data analysis studies drew upon this data source.”

“This data source was also used in one of our other secondary data analysis studies.[12]”

“It was agreed that the findings would be reported across three separate studies (the other two are reported elsewhere[11,12]).”

- 2. It will be very useful for readers to understand the rationale or the author’s reflexivity behind why did the authors pair up the SKINS project and ECO project data to be analysed together. Apart from having project participants from the young person age range, these two projects have diverging project aims and interview schedules, and carried out 3 years apart. Out of all the quotations presented in this submitted manuscript, only 1 quotations was from a ECO project participant. It will be useful for the authors to further reflect on how coding of data from ECO project impacts on coding of SKINS project data, and vice versa, as well as their their impact on the development of themes.**

RESPONSE: We agree that it will be useful to explain the rationale and reflect on the impact that the choice to analyse these two data sets together may have had on the research. Therefore, we have added the following rationale in the data collection section:

“Although both studies had different aims, both used interview questions relevant to this study aim, exploring experiences of eczema treatments and management, and interactions with health professionals.”

We have also added the following reflections in the discussion:

“There are several limitations to this study. Secondary qualitative analysis has been criticised for not considering the socio-cultural-political context under which the primary research was conducted.[24] This issue is particularly salient for this study as the two sets of primary data were collected at two different time points. However, to minimise such decontextualisation, avoid any misinterpreted data, and ensure the final interpretations were representative of both data sets, the interviewers of both data sets were co-authors (DG, AMcN).

Although both primary studies explored the topics relevant to the aims of the current study, these studies were conducted for different purposes (i.e. to inform two different health interventions), which may have introduced differences in the data. For example, sections of the transcriptions and, with participant consent, audio and video clips from the SKINS interviews were to be published online. Therefore, compared to the ECO study participants, SKINS participants may have been more motivated and more likely to give socially desirable answers. Despite this, our analysis highlighted that there was a great deal of convergence between the two data sets and our use of disconfirming case analysis ensured that any differences were identified and acknowledged. Our findings are also congruent with other datasets.[9,10,17,18]”

We have also added in additional quotations and extracts from participants from the ECO study to demonstrate that overall interpretations were made from both datasets and highlighted any divergences between the two data sets.

“One younger participant from the ECO project reported that he found it hard to know how much emollient to apply.”

“I can use one moisturiser for six months, a year, and then my skin will get too used to it and I'll have to keep putting more on. Then I'll end up having to switch to a different kind of moisturiser just so I can get it under control again. (Cassie, female, 15 years old, eczema since birth, ECO project)”

“This support was particularly important for early-mid adolescents whose parents were still involved in their eczema management.

[When I apply treatments depends on] how I'm feeling at the time. So if Mum's at the point of, 'Oh [participant name], go put your creams on', I'm like, 'Okay, I'll go put my creams on' and I'll do it. Then I'll come downstairs and she's double checking, 'Oh have you done your creams?' At times she'd say it and I'll say, 'Yes, I'll go do it and then I'll completely forget,' and that [is] pretty much the days where I would not do them. (Ben, male, 14 years old, eczema for 6 months, ECO project)”

“I normally try to stop, like stop myself from doing it, but it gets to the point where it's just all over, all over my neck and I just have to scratch it. (Amelia, female, 15 years old, eczema for 12 years, ECO project)”

3. It is disappointing to see the data analysis method described in page 9 is so brief and does not adequately explain how the author's thematic analysis was performed, how was the data synthesised and whether the authors followed Braun and Clarke's six step thematic analysis or performed a variation of it. While the SKINS project coding has benefited from member checking, the authors stated KG coded the ECO data, and updated the coding manual accordingly – suggesting the ECO data did not benefit from member checking. Furthermore, Ref 3 and 9 provided a much fuller account of the study methods – the authors could have referenced those papers for full details of their data analysis method performed in relation to the work reported in this manuscript if they did not wish to provide the full details.

RESPONSE: We agree that this manuscript would be strengthened by including additional details of the data analysis method. Therefore, we have added the following detail to the data collection and analysis sections:

“Interviews were either audio or video recorded and transcribed verbatim, and checked by participants for accuracy.”

“Both data sets were analysed together by following the six stages of Braun and Clarke's inductive thematic analysis[14] and data handling was facilitated using NVivo 12 Pro. DG carried out initial coding on the entire SKINS dataset, familiarising herself with the data and generating initial codes that represented the various topics present across the data (e.g. barriers to emollients, experiences with health professionals). A coding manual was created, including descriptions of the codes and example quotes, and discussed with KG and IM (both health psychologists experienced in qualitative research) and MS (GP experienced in qualitative research). These three researchers used the manual to code two transcripts and changes to the coding manual were made following discussions. The revised coding manual was discussed with a PPI representative (AR) and the interviewer for the SKINS project (AMcN). It was agreed that the findings would be reported across three separate studies (the other two are reported elsewhere[11, 12]).

For this study, KG reviewed the initial codes deemed relevant to the study aim, redefined them in line with these aims, searched for, reviewed and defined themes, and created a coding manual for this study. KG then familiarised herself with the ECO data, and coded the data, identifying any data that deviated from the SKINS data (e.g. any new codes or experiences) in the coding manual. Disconfirming cases were identified throughout the entire dataset. Writing up the thematic analysis in this manuscript helped finalise the analysis, enabling the final themes to be reviewed and agreed by all co-authors. Pseudonyms were given to all participants.”

We have referenced the other two secondary data analysis studies, but not for providing more details about the analysis, as the data analysis method was slightly different for each study.

4. Page 19 study limitations

The limitations of this study has not been adequately discussed. The sample size of 28 interviews is at best a modest sample and data saturation was not achieved. The authors should also mention or reference the limitations they stated in Ref 3 and Ref 9 (due to reasons already mentioned under Major Comment 1).

RESPONSE: We agree that this manuscript will benefit from a discussion of the study sample size and data saturation and the inclusion of additional limitations from References 3 and 9. Therefore we have added the following to the methods and limitations section:

“An important quality criteria in qualitative research is the adequacy of the sample size. This can be assessed by consideration of a study's ‘information power’ [15] and whether data saturation was reached.[16] Drawing on the guidelines on information power in qualitative interview studies,[15] our sample size for the secondary data analysis (n=28) was deemed

adequate given the specificity of the study aim (self-management) and target group (young people), the large amount of data (in particular from the SKINS interviews which lasted up to two hours), and the likely high quality of dialogue from using experienced qualitative postdoctoral researchers. It was also comparable to other qualitative interview studies exploring experiences among people with eczema[10]. When analysing the SKINS data, we felt that data saturation was reached as the later interviews did not identify any additional codes or diversely different experiences or views.[16] Although the analysis of the ECO project data did not identify any additional codes (i.e. code saturation was reached [16]), the analysts felt that additional interviews with this age group was needed to be confident that the issues facing this group are fully understood (i.e. meaning saturation was not reached for this group[16]).”

“As this was a secondary data analysis, it was not possible to carry out additional interviews in order to have full confidence that data saturation was reached. Moreover, the sample included only eight males. Additional interviews with adolescents aged 13-16 and males with eczema additional may have enabled us to explore any differences between the age groups and between genders in more depth.”

Also see the limitations included to address comment 2.

- 5. Introduction page 5 line 16 stated ‘Eczema management focuses on the regular use of emollients to retain the skin’s barrier function; treating flare-ups using topical corticosteroids (TSC); identification and avoidance of irritants/triggers; and avoiding scratching, which can further exacerbate eczema symptoms.[4]’ Comments: This statement oversimplified the management of eczema, in contrast to the study findings that eczema treatment is complex for young people. Also, this is not a full description on the management of eczema. While the author’s statement above is relevant to all severity of eczema, however, as the authors referenced the NICE guideline (Ref 4) they should considering further mentioning topical calcineurin inhibitors, bandages, systemic therapy and phototherapy for moderate and severe eczema.**

RESPONSE: We have added the following detail regarding eczema management for those with moderate and severe eczema:

“For most, eczema management focuses on the regular use of emollients to retain the skin’s barrier function; treating flare-ups using topical corticosteroids (TSC); identification and avoidance of irritants/triggers; and avoiding scratching, which can further exacerbate eczema symptoms.[6] For those with moderate and severe eczema, management may also involve use of topical calcineurin inhibitors, bandages or medicated dressings, systemic therapy (immunosuppressants), and phototherapy (light therapy).[6]”

- 6. Results section page 9-10 ‘This led to them to change emollients or TCS, apply increasingly more treatments to get the same effect, try a stronger TCS, or try a different type of treatment (eg. Phototherapy, immunosuppressants).**

Comment: Dermatologists always make the effort to minimise TCS skin thinning risks by introducing steroid sparing agents like topical calcineurin inhibitors whenever indicated. Primary care has very good awareness of these steroid sparing agents and often initiated them prior to secondary care referral. Many young people that I care for with eczema can recall the name of the topical calcineurin inhibitor and know how to use it. Thus, I was very surprised that topical calcineurin inhibitors (Elidel cream and Protopic ointment) was not mentioned in the study findings at all. To me as a dermatologist, it is a significant finding that there is a lack of mentioning/awareness of topical calcineurin inhibitors amongst young persons which is second line treatment for eczema (before considering phototherapy or systemic immunosuppressants). I noted the authors’ COREQ statement stated the study did not aim for saturation of data.

RESPONSE: Only three participants specifically reported their experiences with topical calcineurin inhibitors. Most participants talked about their topical treatments in general terms, rather than naming specific treatments. However, we agree that it is important to also include any relevant experiences of using topical calcineurin inhibitors, therefore, we have added in the following detail:

“Generally, young people believed that their topical treatments (emollients and topical corticosteroids [TCS], topical calcineurin inhibitors) were effective.”

“Two participants described how they had experienced side-effects (red, heated, and stinging skin) from using a topical calcineurin inhibitor.”

“Willow (female, 23 years old, eczema since birth, SKINS project) explained how she asked her GP if she could be given a topical calcineurin inhibitor after a friend with eczema had recommended it. Although her GP prescribed it for her, she felt frustrated that she had to find out about this herself and the GP had not recommended it to her earlier.”

On re-reviewing the data, we were unable to establish the exact order in which the participants tried the various treatments and whether they tried topical calcineurin inhibitors before going on to use phototherapy and immunosuppressants. Therefore, we have removed the last part of the sentence mentioned by the reviewer to avoid confusion. This now reads: *“This led them to change emollient or TCS, apply increasingly more treatment to get the same effect, or try a stronger TCS.”*

7. Introduction page 5 line 10 stated

Although eczema typically starts in infancy and is expected to resolve by late childhood, for many, symptoms can persist into adolescence and adulthood.[3] Comments: the sentence appeared ambiguous. In practice, roughly a quarter of children with eczema may find their eczema persist through to adulthood. Thus, I would be grateful if you could clarify the phrase ‘expected to resolve by late childhood’ by whom? Is it parents, the young person, the clinician or the society?

RESPONSE: We agree that clarification is needed here. Therefore, the sentence now reads:

“Although eczema typically starts in infancy and epidemiology studies show that it typically improves or resolves by late childhood,[3,4] for many, symptoms can persist into adolescence and adulthood.[5]”

8. For all quotations, could you kindly add in whether the quote was from a participant in the SKINS or ECO project please.

RESPONSE: We have made the suggested change throughout.

9. Regarding Table 1 in this manuscript. It looks the same as the Table 1 in Ref 9. Consider adding a reference please.

RESPONSE: Although the table looks the same as that in Reference 9 and is summarising data for the same sample, it is slightly different as the table in this study includes details on ethnicity. Therefore, it would not be appropriate to reference the other table.

10. Reference 9 has been published and the authors should update this reference

RESPONSE: We have now updated this reference.

11. The supplementary files 1 and 2 (Interview schedule for SKINS project and ECO Project are almost the same in this submitted manuscript and in Ref 9, but please pay attention to the differences in Supplementary 2 Interview Schedule – Eczema Care Online Project (see below) and make the necessary corrections as the interview schedules should be exactly the same.

In this submitted version supplementary file 2, page 4 ends with That is all really useful, thank you. Is there anything that we haven't talked about that you would like to add?

In Ref 9 supplementary file 2, page 4 ends with Views about wording – extracts Show the extract – That is all really useful, thank you. Is there anything that we haven't talked about that you would like to add?

RESPONSE: The 'Views about the wording' section mentioned in the interview schedule in Reference 9 was part of a separate think-aloud interview study and the data for this was not part of the ECO dataset used in this analysis. Therefore, we removed this section from the interview schedule in this manuscript to avoid confusion.

Reviewer 2 comments:

1. In the introduction TSC is written instead of TCS 2. on page 6, third paragraph, the wording of "from 8 months to all their life" was a bit unclear, I initially read it as from 8 months of age instead of 8 months in total. Could be clearer if written something like "had eczema for a period of 8 months to all their life"

RESPONSE: We agree that this statement is unclear and have amended it as suggested.

2. there is inconsistency with TCS and TCSs both being used for the plural "topical corticosteroids" throughout the paper - eg in the introduction and last paragraph of page 8

RESPONSE: We have now changed all instances to TCS for consistency.

VERSION 2 – REVIEW

REVIEWER	Dr Simon Tso Jephson Dermatology Centre, South Warwickshire NHS Foundation Trust, Warwick, United Kingdom
REVIEW RETURNED	02-Dec-2020

GENERAL COMMENTS	Thank you for taking the time to revise the manuscript in line with my previous recommendations. The methodology section has been strengthened, a fuller discussion on the study limitations has been provided, and a fuller account of the authors' reflexivity on the subject has been included. I am satisfied with the changes made to the manuscript. A further point for all health professionals to reflect on, after reading your manuscript again, is the cost of treatments which you have highlighted as a barrier to self management of eczema. From my experience, many adult patients, especially young adults, are not aware of the prescription prepayment certificate which can help to control their treatment costs. Signposting them to the prescription prepayment certificate may help to address the financial burden of treatments.
---